# Teleporter Theory: A General and Simple Approach for Modeling Cross-World Counterfactual Causality

## Abstract

Leveraging the development of structural causal model (SCM), researchers can establish graphical models for exploring the causal mechanisms behind machine learning techniques. As the complexity of machine learning applications rises, *single-world* interventionism causal analysis encounters theoretical adaptation limitations. Accordingly, *cross-world* counterfactual approach extends our understanding of causality beyond observed data, enabling hypothetical reasoning about alternative scenarios. However, the joint involvement of cross-world variables, encompassing counterfactual variables and factual variables, challenges the construction of the graphical model. Existing approaches, e.g., Twin Network and Single World Intervention Graphs (SWIG), establish a symbiotic relationship to bridge the gap between graphical modeling and the introduction of counterfactuals albeit with room for improvement in generalization. In this regard, we demonstrate the theoretical *limitations* of certain current methods in cross-world counterfactual scenarios. To this end, we propose a novel *teleporter theory* to establish a general and simple graphical representation of counterfactuals, which provides criteria for determining *teleporter* variables to connect multiple worlds. In theoretical application, we determine that introducing the proposed teleporter theory can directly obtain the conditional independence between counterfactual variables and factual variables from the cross-world SCM without requiring complex algebraic derivations. Accordingly, we can further identify counterfactual causal effects through cross-world symbolic derivation. We demonstrate the generality of the teleporter theory to the practical application. Adhering to the proposed theory, we build a plug-and-play module, and the effectiveness of which are substantiated by experiments on benchmarks.

## 1 Introduction

Causal inference is a specialized field that presents promising potential with respect to improving machine learning methods, conventionally encompassing four steps: 1) causal model construction for modeling causality in machine learning applications in a qualitative analysis manner (Liu et al., 2021; Chen et al., 2022; Li et al., 2023b); 2) causal model validation, including independence and causality testing, to demonstrate the correctness of the causal model construction (Daniusis et al., 2012; Zhang et al., 2012; Lu et al., 2021); 3) causal model-based deconfounding approach implementation, which prospers in various machine learning fields, e.g., eliminating spurious correlations (Mao et al., 2021; Liu et al., 2022a) and performing counterfactual reasoning (Chang et al., 2021) in computer vision, learning the intrinsic rationale of the graph (Ji et al., 2024; Wu et al., 2024) in graph neural networks, overcoming selection bias (Li et al., 2023a) and popularity bias (Zhao et al., 2022) in the recommendation systems; 4) deconfounding approach estimation improvement, focusing on enhancing the accuracy of causal model-based deconfounding (Frauen et al., 2023; Zhu et al., 2024). Benefiting from the establishment of graphical models, the advances of the structural causal model (SCM) concentrate greater potential onto exploring the causal mechanisms behind machine learning techniques, e.g., the analysis of independent relationships among variables and the identification of causal effects for various machine learning applications.

In practice, the involvement of derived discrete data with extra stringent structural constraints increases the complexity of machine learning application scenarios, resulting in a lack of adaptability of conventional interventionism causal analysis theories, i.e., *single-world* SCM-based theory (Xia et al., 2021; Zečević et al., 2021; Pawlowski et al., 2020). *cross-world* counterfactuals (Correa et al., 2021; Richens et al., 2022; Shah et al., 2022; Alomar et al., 2023) provide a framework to estimate "what-if" scenarios that transcend the observed world, aiding in better-informed causal inference, which is crucial for understanding causal relationships in a more comprehensive manner, as it enables the exploration of causality under different hypothetical conditions (Shalit et al., 2017; Ibeling & Icard, 2020; Khemakhem et al., 2021; Sanchez-Martin et al., 2021; D'Amour et al., 2022). A focal issue is the exclusivity of counterfactual variables and factual variables in an invariant graphical model, challenging the construction of cross-world counterfactual SCMs. In this regard, Twin networks (Balke & Pearl, 1994; Han et al., 2022; Vlontzos et al., 2023) demonstrate a symbiotic relationship of graphical modeling in counterfactual and real-world scenarios. SWIG (Richardson & Robins, 2013; Hernán & Robins, 2020) presents a simple graphical theory unifying causal directed acyclic graphs (DAGs) and potential (aka counterfactual) outcomes for identifying counterfactual queries. Yet, in this paper, we provide multiple scenarios of cross-world counterfactual causal analysis, where the applications of representative approaches are *limited*, detailed in Section 3.

To this end, we propose the *teleporter theory* to establish a complete graphical representation of counterfactuals, providing a general and simple approach for modeling cross-world counterfactual causality. Concretely, according to the framework of probabilistic causal models, each variable can ultimately trace its changes back to the exogenous nodes that influence it by iteratively applying the structural equations of its parent nodes over a finite number of steps. Variables that have consistent structural equations in both the real world and the counterfactual world possess equivalence, which is determined as a *teleporter*, and thus we can construct cross-world SCM by using the teleporter variables. Accordingly, we provide sufficient causal analysis from the *structural equation* perspective, substantiating the theoretical correctness of the proposed teleporter theory. In terms of theoretical applications, we focus on two main aspects: 1) we apply $d$-separation to test the conditional independence between any two cross-world variables of a cross-world SCM constructed by introducing the teleporter theory, which can prove the correctness and generalization of our theory; 2) we use the teleporter theory to build the cross-world SCM and further leverage the cross-world symbolic derivation to compute counterfactual probability, which can avoid the complex calculation of the probability distribution of background variables, demonstrating the effectiveness and simplicity of our theory. Adhering to the proposed theory, we build a practical plug-and-play module to address the intrinsic issue in the field of Graph Out-Of-Distribution (GraphOOD) (Gui et al., 2022; Chen et al., 2022; Jia et al., 2024). The consistency and effectiveness of the proposed module are substantiated by experiments on benchmarks.

Our **contributions** are as follows: (1) We provide multiple motivating examples to elucidate the incompleteness of current approaches in certain cross-world counterfactual scenarios with sufficient causal analysis. (2) We propose a general and simple approach for modeling cross-world counterfactual causality, namely the teleporter theory, which is proved as a complete causal analysis method. (3) We provide sufficient evidence to prove the theoretical correctness of the proposed teleporter theory by introducing the structural equation analysis. (4) We conduct extensive validations on commonly adopted benchmarks, demonstrating the generalized applicability of the teleporter theory from theoretical and practical perspectives.

## 2 RELATED WORK

**Modeling Single-World Causality.** Single-world interventions pertain to the first two levels of Pearl's causal hierarchy (Pearl, 2009b; Bareinboim et al., 2022): association and intervention. Once we can model causality using observational data (Perkovi et al., 2018; Jaber et al., 2019), various methods exist for estimating interventional distributions (Kocaoglu et al., 2017; Ke et al., 2019; Xia et al., 2021; Zečević et al., 2021), provided identifiability is ensured (Bareinboim et al., 2022). The implementation of interventions transcends simple modeling of data associations, aiming instead to answer scientific questions such as "How effective is $X$ in influencing $Y$?" and thus achieving estimates of causal effects. Numerous works in the machine learning community have benefited from this approach: (1) real user preference in recommendation systems, such as deconfounding (Zhang et al., 2023a;b; He et al., 2023) and disentangling (Sun et al., 2022), (2) rationale representations

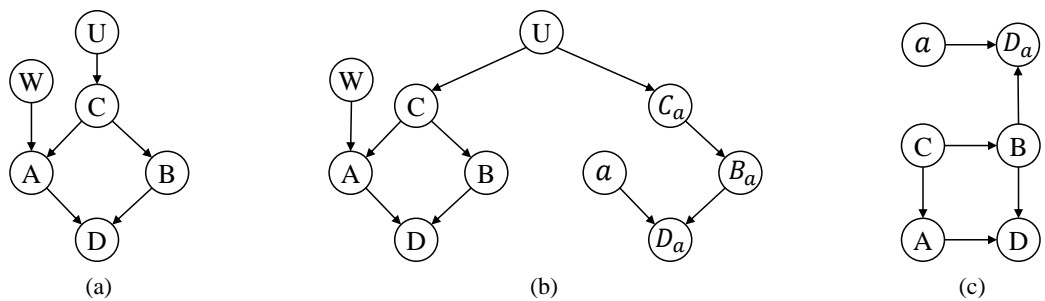

Figure 1: Example for inapplicability of twin network: Figure (a) represents the real-world SCM, Figure (b) shows the cross-world SCM constructed using the twin network, and Figure (c) illustrates the cross-world SCM constructed using the teleporter theory.

in graph neural networks, such as robustness and invariant subgraphs (Chen et al., 2022; Wu et al., 2022), and (3) invariant representations in domain generalization, such as eliminating spurious correlations (Arjovsky et al., 2019; Cui & Athey, 2022).

**Modeling Cross-World Counterfactual Causality.** Cross-world causality aims to address the top-level query of Pearl's causal hierarchy (Pearl, 2009b; Bareinboim et al., 2022): counterfactuals. However, estimating counterfactual causality faces the challenge of conflicts between factual variable values and counterfactual variable values, making identifiability (Ibeling & Icard, 2020; Khemakhem et al., 2021; Geffner et al., 2022; D'Amour et al., 2022) more scarce compared to interventions. Despite this, answering counterfactual queries like "why?" and "what if?" using causal framework can enable personalized and interpretable decision-making and reasoning. This significantly advances several key areas: 1) application in computer vision, e.g., alleviating data scarcity through data augmentation (Kaushik et al., 2019; Xia et al., 2022); 2) fairness in legal and policy-making contexts (Kusner et al., 2017; Zhang & Bareinboim, 2018); 3) interpretability in the field of medical health (Oberst & Sontag, 2019; Richens et al., 2022), among others.

## 3 LIMITATIONS OF EXISTING CROSS-WORLD GRAPHICAL MODELS

Before formally introducing our proposed new graphical model, we reviewed existing classical and widely influential cross-world graphical models for estimating counterfactual causality: Twin Network (Balke & Pearl, 1994) and Single World Intervention Graphs (SWIG) (Richardson & Robins, 2013). In this section, we provide detailed case studies to analyze the limitations and constrained scenarios of the existing cross-world graphical models.

### 3.1 THEORETICAL INAPPLICABILITY OF TWIN NETWORK

Twin network (Balke & Pearl, 1994) is formed to model cross-world counterfactual causality by connecting the real world and counterfactual world, sharing exogenous variables between them. The constructed sub-networks of real world and counterfactual world are structurally identical, except that the arrows pointing to the intervened variable are removed in the counterfactual sub-network. The specific construction steps are as follows: 1) duplicating the endogenous variables $X = \{X_1, X_2, ..., X_n\}$ from the real world as endogenous variables $X^* = \{X_1^*, X_2^*, ..., X_n^*\}$ in the counterfactual world; 2) selecting the intervened variable $X_i$ and removing all arrows pointing to the counterfactual variable $X_i^*$; 3) connecting $X$ and $X^*$ through existing exogenous variables $U$ to form the twin network. Fig. 1(b) illustrates an example of cross-world SCM constructed by using twin network, where the intervened variable is $A$, the value of the counterfactual variable $A^*$ is $a$, and the existing exogenous variables are only $U$ and $W$. We also provide an analysis in the **Appendix** B.2 for the example that constructs twin network using all exogenous variables.

However, the benchmark twin network encounters *theoretical inapplicability* in certain scenarios of modeling cross-world counterfactual causality. Concretely, we explore the firing squad example in "*Causality*" (Pearl, 2009b) p. 213, Fig. 7.2, as depicted in Fig. 1(a) and (b). We aim to test whether $D_a$ is independent of $A$ given $B$ or $C$, i.e., $A \perp\!\!\!\perp D_a \mid B$ or $A \perp\!\!\!\perp D_a \mid C$. The corresponding twin network of this example causal graph is illustrated in Fig. 1(b). To assess the conditional

independence between $A$ and $D_a$, we determine under which variables $A$ and $D_a$ are $d$-separated. Conditional on $C$, the path from $A$ to $D_a$, i.e., $A \leftarrow C \leftarrow U \rightarrow C_a \rightarrow B_a \rightarrow D_a$, is blocked by node $C$, and thus, $A \perp\!\!\!\perp D_a \mid C$ holds. Yet, conditional on $B$, this path is $d$-connected, i.e., $A \not\perp\!\!\!\perp D_a \mid B$.

We validate the conclusions obtained from the twin network from two perspectives: 1) empirical conclusions from (Pearl et al., 2016) Theorem 4.3.1 (Counterfactual Interpretation of Backdoor) on p. 102 and 2) the quantitative analysis by introducing a numerical example. According to Theorem 4.3.1, both variables $B$ and $C$ satisfy the back-door criterion for $(A, D)$, indicating that for all values $a$ of $A$, given $B$ or $C$, the counterfactual $D_a$ is conditionally independent of $A$.

On the other hand, considering the firing squad example in Fig. 1(a), $A$ and $B$ are the officers, $C$ is the captain (waiting for the court order $U$), and $D$ represents the condemned prisoner. The exogenous variables are only $U$ and $W$, which represent the court order and the nervousness of police officer $A$, respectively. The values and meanings of each variable are as follows:

1. $A(u, w), B(u, w)$ indicate whether officers $A$ and $B$ fire their guns, respectively, and $D(u, w) = 1$ indicates the death of the prisoner. The prisoner will not die from any other factors besides the executioners, so we ignore the exogenous variables for $D$.

2. $D_0(u, w)$ and $D_1(u, w)$ represent the counterfactual values under interventions $A = 0$ and $A = 1$, respectively.

3. $P(u = 1) = p$ represents the probability of issuing a death sentence, $P(w = 1) = q$ represents the probability that officer $A$ pulls the trigger due to nervousness. For the specific values of the variables, please refer to Table 2 in **Appendix** B.1.

Verify that $\mathbf{D_a} \not\perp\!\!\!\perp \mathbf{A}$: $P(D_0 = 1) = p, P(A = 1) = 1 - (1 - p)(1 - q), P(D_0, A = 1) = p$. Therefore, $P(D_0, A = 1) = p \neq p(1 - (1 - p)(1 - q)) = P(D_0 = 1)P(A = 1)$.

Verify that $\mathbf{A} \perp\!\!\!\perp \mathbf{D_a}|\mathbf{B}$: $P(D_0 = 1|B = 1) = 1, P(D_0 = 1|B = 1, A = 1) = 1, P(D_0 = 0|B = 1) = 0, P(D_0 = 0|B = 1, A = 1) = 0$. The remaining values can be verified, so $P(D_a|B) = P(D_a|B, A)$. For detailed calculations, please refer to the **Appendix** B.1.

The above analysis demonstrates that the twin network erroneously determines $A \not\perp\!\!\!\perp D_a \mid B$, which contradicts the **actual condition** $A \perp\!\!\!\perp D_a|B$, proving that the twin network lacks theoretical completeness in specific cross-world SCMs.

## 3.2 Deficiencies of Single World Intervention Graphs

Establishing conditional independencies of counterfactuals in graphical models is a significant area of research. A prominent example is the Single World Intervention Graph (SWIG) (Richardson & Robins, 2013), which possesses this property and is covered extensively in relevant textbooks (Hernán & Robins, 2020, Ch. 6). However, we identified a subtle limitation of the SWIG model: when conditioning on certain factual variables of interest, it does not intuitively reveal independence relations from the graphical model. For instance, in Fig. 9 of the **Appendix** C, while the SWIG model can derive $X \perp\!\!\!\perp Y(x)|L_1$ and $X \perp\!\!\!\perp Y(x)|L_1, L_2(x)$, it fails to capture that $X \not\perp\!\!\!\perp Y(x)|L_1, L_2$.

Additional comparison is presented in the **Appendix** C with three examples in Fig. 8–10, demonstrating that our theory which is detailed in Section 4 can indeed construct a complete cross-world graphical model: 1) Consistency with SWIG: Both provide a new graphical view of the back-door formula, yet the twin network has a counterexample; 2) Superiority over SWIG: SWIG cannot intuitively display **all variables in a single graph**, making it difficult to encompass the conditional independence relationships of all variables in both the real world and counterfactual scenarios.

## 4 Teleporter Theory for Modeling Cross-World Counterfactual Causality

To remedy the mentioned theoretical deficiency, we employ a probabilistic causal model framework (Pearl, 2009a) to expound the teleporter theory. As definitions in **Appendix** A, for SCM $M = \langle X, U, F \rangle$, a probabilistic causal model is a tuple $\langle M, P(u) \rangle$, where $P(u)$ is the probability distribution over the set $U$. By the definition of structural equation $x_i = f_i(pa_i, u_i)$, the value of an endogenous variable $X_i$ can be recursively represented by all possible values of exogenous variables,

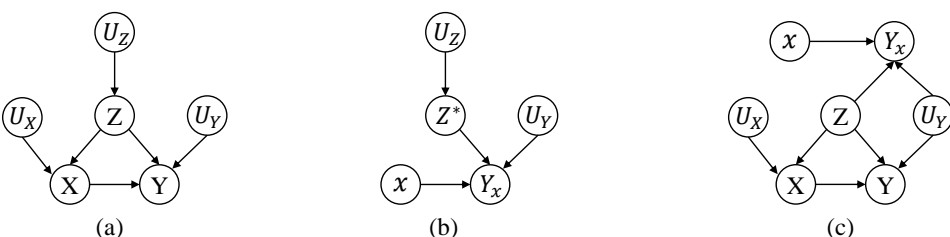

Figure 2: Illustration of cross-world SCM construction using teleporter theory: Figure (a) represents the real world $\mathcal{W}_r$, Figure (b) depicts the counterfactual world $\mathcal{W}_c$, and Figure (c) shows the cross-world SCM $\mathcal{W}_m$ formed by connecting the variables in $\mathcal{W}_r$ and $\mathcal{W}_c$ through the teleporter $Z$.

i.e., $(u_1, u_2, ..., u_n)$, meaning each endogenous variable is a function of $U$. For instance, for a certain endogenous variable $X_i \in X$, we have $P(X_i = x_i) = \sum_{\{u|X_i(u)=x_i\}} P(u)$. We formalize the above assertion, and the value of $X_i$ can be represented by the following recursively defined function:

$$x_i = f_{X_i}(pa_i, u_i) \tag{1}$$
$$= f_{X_i}(f_{X_{i_1}}(pa_{i_1}, u_{i_1}), f_{X_{i_2}}(pa_{i_2}, u_{i_2}), ..., f_{X_{i_k}}(pa_{i_k}, u_{i_k}), u_i) \tag{2}$$
$$= g_{X_i}(u_1, u_2, ..., u_n) \tag{3}$$

where $pa_i = \{X_{i_1}, X_{i_2}, ..., X_{i_k}\} \subset X$, and $g_{X_i}$ is a function determined solely by the exogenous variables $(u_1, u_2, ..., u_n)$ after a finite number of iterations.

Accordingly, we consider the structural equations of variables in the counterfactual world. Suppose the counterfactual world with the intervention $do(X_i = x^\star)$, and the values of variables $X_j$ are determined as follows:

$$x_j = f_{X_j}^{x^\star}(pa_j, u_j) = \begin{cases} x^\star, & X_j = X_i, \\ f_{X_j}(u_j), & X_j \neq X_i \text{ and } pa_j = \varnothing, \\ f_{X_j}(f_{X_{j_1}}^{x^\star}(pa_{j_1}, u_{j_1}), ..., f_{X_{j_k}}^{x^\star}(pa_{j_k}, u_{j_k}), u_j), & \text{otherwise.} \end{cases} \tag{4}$$

According to twin network, the real world and the counterfactual world share only the exogenous variable $U$, while the endogenous variables differ, i.e., $X_i \neq X_i^*$. However, in fact, when comparing the factual value (equation 2) and the counterfactual value (equation 4), there are still certain endogenous variables that have the same values in both the real and counterfactual worlds, such as non-intervened variables without endogenous parent nodes as shown in the second line of equation 4. More generally, variables in the counterfactual world can be classified into two categories, with one class having the same values as in the real world, while the other class has values that differ from those in the real world.

**Lemma 1.** *(Categories of counterfactual variables) Suppose the counterfactual world with the intervention $do(X = x)$, and by removing all arrows pointing to the intervened variable $X$, we obtain the counterfactual causal graph $\mathcal{G}_x$. The counterfactual variables can be divided into two categories: 1) In $\mathcal{G}_x$, the set of descendants of the intervention variable $X = x$, denoted as $D^*$; 2) In $\mathcal{G}_x$, the set of variables $d$-separated from the intervention variable $X = x$, denoted as $Z^*$. The values of the variables in set $Z^*$ remain the same as in the real world, i.e., $Z^* = Z$. In contrast, the values of the variables in set $D^*$ differ from those in the real world. Hence, we denote $D^*$ as $D_x$ to indicate that its values differ from those in the real world.*

For the convenience of counterfactual notation, we present our teleporter theory by introducing endogenous variables as uppercase letters[1] $X, Y, ..., Z$. Accordingly, we provide the detailed theoretical description of our theory.

**Definition 1.** *(Teleporter and merging operation) A pair of variables that have the **same values** in both the real world and the counterfactual world: $Z \leftarrow U_Z \rightarrow Z^*$. We implement a **merging operation** on these pairs of variables (in the cross-world SCM graph, this is represented by merging three variables into one variable $Z$), thus $Z$ is called the **Teleporter**.*

Pearl's twin network fundamentally maintains the invariance of values between the real world and the counterfactual world—specifically, the shared exogenous variables. We have demonstrated that

---

[1]Please refer to Appendix A for the definition of counterfactual notation.

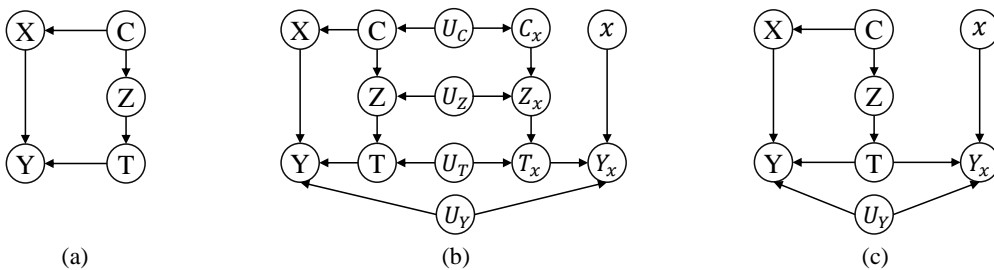

Figure 3: Figure (a) represents the real world $\mathcal{W}_r$, Figure (b) shows the cross-world SCM $\mathcal{W}_m$ constructed using the twin network, and Figure (c) depicts the cross-world SCM $\mathcal{W}_m$ constructed using the teleporter theory.

it is not just the exogenous variables that remain invariant; the Teleporter $Z$ also conforms to this property and can therefore be shared. Please refer to **Appendix** E for the corresponding proofs.

**Theorem 1.** *(Teleporter theory for modeling cross-world counterfactual causality) Suppose we intervene on the endogenous variable $X$. Let $\mathcal{W}_r = \langle M, \mathfrak{u} \rangle$ denote the real world before the intervention, and $\mathcal{W}_c = \langle M_x, \mathfrak{u} \rangle$ denote the counterfactual world, where $\mathfrak{u}$ denotes the variable of the corresponding exogenous variable set $U$, and $\mathfrak{u}$ is shared between $\mathcal{W}_r$ and $\mathcal{W}_c$. $\mathcal{W}_r$ and $\mathcal{W}_c$ can be connected to form a cross-world SCM $\mathcal{W}_m$ by adhering to the following rules:*

- **Rule 1**: *In the counterfactual world $\mathcal{W}_c$, the endogenous variables $Z$, that are **d-separated** from $X = x$, can be determined as the **teleporter**. $\mathcal{W}_m$ is obtained by connecting $\mathcal{W}_r$ and $\mathcal{W}_c$ through the teleporter $Z$.*

- **Rule 2**: *All descendants of $X$ in $\mathcal{W}_r$, e.g., $D$, have the potential value influenced by the intervention $do(X = x)$ in $\mathcal{W}_c$, e.g., $D_x$.*

- **Rule 3**: *The exogenous variable $U_Z$ associated with the teleporter $Z$ is removed, and the teleporter $Z$ is introduced via the Merging Operation to connect $\mathcal{W}_r$ and $\mathcal{W}_c$. The exogenous variable $U_D$ associated with $D_x$ is retained to connect $D \leftarrow U_D \rightarrow D_x$.*

We illustrate the implementation process of teleporter theory by using a classic causal graph as an example. Fig. 2(a) investigates the causal relationship between $X$ and $Y$, where $Z$ acts as a confounder (Pearl, 2009b), and all exogenous variables are depicted in the graph, which is treated as the real-world SCM $\mathcal{W}_r$. Fig. 2(b) represents the intervention $do(X = x)$ on $X$, where all arrows pointing to $X$ are removed, which is treated as the counterfactual world $\mathcal{W}_c$. Next, we clarify the variables in the counterfactual SCM $\mathcal{W}_c$. According to *Rule 1* of Theorem 1, in Fig. 2(b), the only endogenous variable that is $d$-separated from $X = x$ is $Z^*$, with its structural equation denoted as $f_{Z^*}(u_Z)$. $X$ is *not* a parent node of these variables, and therefore they are not influenced by the intervention $do(X = x)$. The value of $Z^*$ in the counterfactual world equals that of $Z$ in the real-world graph in Fig. 2(a), thus it is called the teleporter. Furthermore, we determine that the value of structural equation for $Y$ in $\mathcal{W}_c$, denoted as $f_{Y_x}(X = x, Z, u_Y)$, is clearly different from the value of structural equation for $Y$ in $\mathcal{W}_r$, denoted as $f_Y(X, Z, u_Y)$. Therefore, the meaning and value of $Y$ are different in the two worlds. According to *Rule 2* of Theorem 1, the descendants of $x$ consist only of $Y$ in $\mathcal{W}_c$, denoted as $Y_x$. For further discussion on the meaning of factual and counterfactual variables, please refer to the additional analysis in **Appendix** D.1.

To derive the cross-world SCM, which connects the real world and counterfactual world, we introduce the teleporter theory. According to *Rule 3* of Theorem 1, the exogenous variable $U_Z$ associated with the teleporter $Z$ is removed, and the connecting path between $\mathcal{W}_r$ and $\mathcal{W}_c$, $X \leftarrow Z \leftarrow U_Z \rightarrow Z^* \rightarrow Y_x$, is merged into $X \leftarrow Z \rightarrow Y_x$. In addition, the exogenous variable $U_Y$ associated with $Y_x$ is retained, connecting $Y \leftarrow U_Y \rightarrow Y_x$. By utilizing the common teleporter $Z$ shared between $\mathcal{W}_r$ and $\mathcal{W}_c$ as a connecting node along with the remaining exogenous variables, we derive the cross-world SCM $\mathcal{W}_m$ depicted in Fig. 2(c).

## 5 THEORETICAL APPLICATION OF TELEPORTER THEORY

The SCM $M$ and its corresponding graph $\mathcal{G}$ facilitate the graphical representation of causal variables, enabling us to intuitively test the independence between variables ($d$-separation). This, in turn, allows

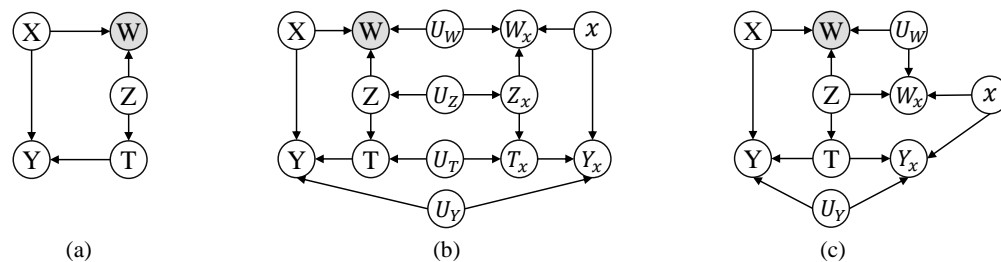

Figure 4: Figure (a) represents the real world $\mathcal{W}_r$, Figure (b) shows the cross-world SCM $\mathcal{W}_m$ constructed using the twin network, and Figure (c) depicts the cross-world SCM $\mathcal{W}_m$ constructed using the teleporter theory. Grey nodes indicate conditioning on that variable

us to explore the effects of interventions without conducting new experiments, e.g., back-door/front-door adjustments. However, counterfactual variables $Y_x$ and factual variables $X$ cannot coexist in a single graph $\mathcal{G}$ due to involving cross-world considerations. Twin network (Balke & Pearl, 1994) is the first attempt to address this issue, yet such a method fails in certain scenarios, shows its theoretical incompleteness. In the following two subsections, we demonstrate that the teleporter theory can provide a complete graphical representation of counterfactuals.

### 5.1 INDEPENDENCE BETWEEN CROSS-WORLD VARIABLES

The cross-world independence between counterfactual variables and factual variables is difficult to derive from the separated real-world and counterfactual SCMs or the corresponding structural equations. The significant advantage of the teleporter theory lies in the graphical representation of counterfactuals, enabling us to analyze the (conditional) independence between any pair of cross-world variables. Concretely, considering the inapplicability of twin network in Section 3.1, we propose to demonstrate the theoretical completeness and generalization of the proposed teleporter theory as follows. In the cross-world SCM $\mathcal{W}_m$ of Fig. 1(c) obtained through the teleporter theory, we conclude that both $A \perp\!\!\!\perp D_a \mid B$ and $A \perp\!\!\!\perp D_a \mid C$ hold. This is because the path from $A$ to $D_a$, i.e., $A \leftarrow C \rightarrow B \rightarrow D_a$, is blocked by $B$ or $C$. Similarly, upon adding $D$ as a condition, new paths between $A$ and $D_a$ are opened through the collider node $D$. Therefore, conditional on $\{D, C\}$, $A$ and $D_a$ are *not* d-separated, satisfying $A \not\perp\!\!\!\perp D_a \mid \{D, C\}$. However, conditional on $\{D, B\}$, $A$ and $D_a$ are d-separated, satisfying $A \perp\!\!\!\perp D_a \mid \{D, B\}$.

Thus, the proposed teleporter theory can widely empower the (conditional) independence testing between cross-world variables. To better demonstrate the implementation of independence testing in the cross-world SCM built by using the teleporter theory, we propose the following Theorem 2, summarizing the d-separation theorem for counterfactuals. The proof can be found in the **Appendix F**.

**Theorem 2.** (*d-separation for cross-world variables under the teleporter theory*) *In the cross-world SCM $\mathcal{W}_m$ constructed by following the teleporter theory, the path $p$ between the factual variable $X$ and the counterfactual variable $Y_x$ is d-separated by the node set $Z$ if and only if:*

*1. $p$ contains either a chain structure or a fork structure, with intermediate nodes in $Z$, or*

*2. $p$ contains a collider structure, with neither the intermediate node nor its descendants in $Z$.*

*The set $Z$ d-separates $X$ from $Y_x$ if and only if $Z$ blocks all paths from $X$ to $Y_x$.*

### 5.2 CROSS-WORLD ADJUSTMENT

The joint distribution of counterfactual statements requires computation, storage, and utilization of the marginal probability of values of the exogenous variables, i.e., $P(u)$. For example, $P(Y_x = y, X = x') = \sum_{\{u \mid Y_x(u)=y, X(u)=x'\}} P(u)$. Classic works summarize three steps for estimating the counterfactual $P(Y_x \mid e)$ in (Pearl, 2009b), where $e$ denote the observed variable values: 1) abduction: updating $P(u \mid e)$ using the fact $e$; 2) action: updating the SCM $M$ to $M_x$; 3) computing $P(Y_x \mid e)$ in the counterfactual world $\mathcal{W}_c = \langle M_x, P(u \mid e) \rangle$. However, obtaining the distribution of exogenous variables $U$ is extremely difficult. The teleporter theory provides a simple method to compute $P(Y_x \mid e)$, facilitating cross-world adjustment.

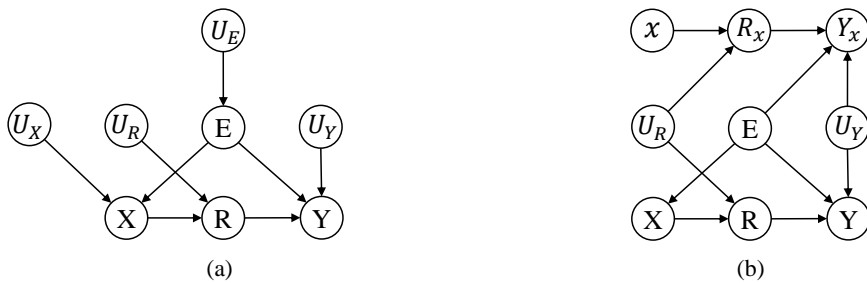

(a)                                  (b)

Figure 5: SCM for GraphOOD. Figure (a) denotes the real-world SCM. Figure (b) denotes the cross-world SCM.

We propose the counterfactual criterion to obtain the (conditional) independence of $X$ and $Y_x$:

**Theorem 3.** *(**Counterfactual criterion and cross-world adjustment**) The evidence $e$ represents the values of the variable $E$ in the real world $\mathcal{W}_r$. Given an observable variable set $Z$, if $E \cup Z$ causes that conditional on $E \cup Z$, $X$ and $Y_x$ are $d$-separated in the cross-world SCM $\mathcal{W}_m$, then the counterfactual $Y_x$ is conditionally independent of $X$, denoted as $X \perp\!\!\!\perp Y_x \mid \{E, Z\}$. The cross-world adjustment formula can be derived as follows:*

$$P(Y_x = y \mid E = e) = \sum_z P(Y = y \mid Z = z, X = x, E = e)P(Z = z \mid E = e). \tag{5}$$

In comparison to the counterfactual interpretation of the back-door criterion in (Pearl et al., 2016), our theoretical approach is proved to be a generalized solution, since the former approach can only treat the back-door path-related scenarios, our approach can achieve cross-world adjustment for any pair of variables. Please refer to **Appendix** G for the corresponding proofs.

We will now present two examples to illustrate that the teleporter theory is more complete compared to twin network, as the latter fails in multiple scenarios. The first example demonstrates cases where twin network *incorrectly* identifies the required variables for adjustment. The SCM of such an example is depicted in Fig. 3(a). In the twin network of Fig. 3(b), $X$ and $Y_x$ are $d$-connected by the path $X \leftarrow C \leftarrow U_c \rightarrow C_x \rightarrow Z_x \rightarrow T_x \rightarrow Y_x$. If we acquire to compute $Y_x$, the variables for adjustment can only be $C$, since using $Z$ or $T$ for adjustment would open up a collider node, resulting in certain dependence relationships between the parent nodes. However, in the cross-world SCM $\mathcal{W}_m$ of Fig. 3(c) modeled by using the teleporter theory, the path between $X$ and $Y_x$ is $X \leftarrow C \rightarrow Z \rightarrow T \rightarrow Y_x$. According to Theorem 3, we can perform the adjustment on any node in $\{C, Z, T\}$, which is consistent with the empirical conclusion in "*Causality*" (Pearl, 2009b).

The second example, as illustrated in Fig. 4(a), involves computing $P(Y_x \mid w)$ given the known evidence $w$. In the twin network of Fig. 4(b), $X$ and $Y_x$ are connected only through one path: $X \rightarrow \underline{W} \leftarrow Z \leftarrow U_z \rightarrow Z_x \rightarrow T_x \rightarrow Y_x$[2]. In this case, we can only perform the adjustment on $Z$, since the adjustment on $T$ would open up new paths, i.e., $Z \not\perp\!\!\!\perp U_T \mid T$. The cross-world SCM $\mathcal{W}_m$ constructed by using the teleporter theory is depicted in Fig. 4(c), and according to Theorem 3, we can perform the adjustment on both $T$ and $Z$, which well fits the empirical conclusion in "*Causality*" Pearl (2009b). The above theoretical application analysis sufficiently demonstrate the generalization and applicability of our teleporter theory.

In addition, we conducted a deeper analysis of the theoretical applications of the teleporter theory in the **Appendix** D, demonstrating: 1) How to obtain conditional exogeneity to control for confounding bias and identify the correct adjustment variables. For example, when calculating $P(Y_{x'}|x, y)$ or $P(Y_{x'}|y)$, we need to examine the conditional independence of given factual variables $Y$ to determine the appropriate adjustment variables. The twin network, however, incorrectly selects the adjustment variables, making it difficult to control for confounding bias. For specific numerical examples, please refer to the **Appendix** D.2; 2) The significant potential of teleporter theory in computing complex counterfactual queries.

---

[2] $\underline{W}$ represents conditioning on W, i.e., W is given.

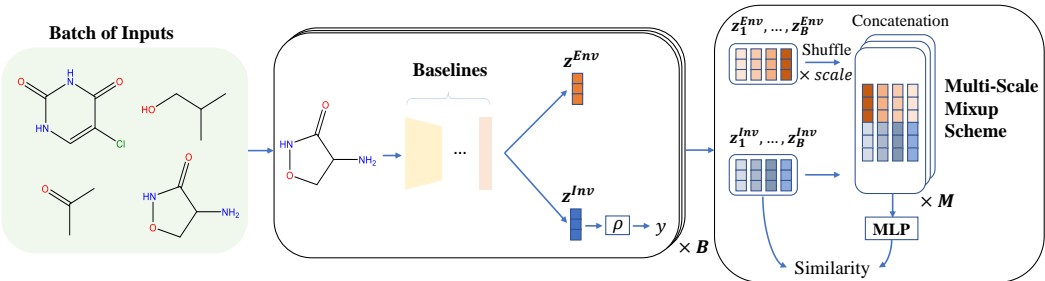

Figure 6: GraphOOD learning paradigm with Multi-Scale Environment Mixup Scheme.

## 6 PRACTICAL APPLICATION OF TELEPORTER THEORY

To demonstrate the practical applicability of teleporter theory, we utilize our theory to model the set of variables in GraphOOD (Gui et al., 2022; Chen et al., 2022; Jia et al., 2024), and propose a method to perform the counterfactual conditional probability estimation between cross-world variables.

**Preliminary of GraphOOD.** The area of OOD learning deals with scenarios in which training and test data follow different distributions. Furthermore, GraphOOD problems focus on not only general feature distribution shifts but also structure distribution shifts (Gui et al., 2022). Accordingly, Graph neural networks (GNNs) (Kipf & Welling, 2016; Xu et al., 2018) are designed based on node features and adjacent matrix to pass messages, which perform well in solving GraphOOD problems. Previous GNN-based methods (Yang et al., 2022; Chen et al., 2022; Zhuang et al., 2024) aim at disentangling the invariant part and the environment part of the input graph to find rationales, thereby addressing the problem of domain shift. The common learning paradigm is shown in Fig. 6, where the input graph is processed and split into two latent variables, i.e., the invariant representation $z^{Inv}$ and the environment-dependent representation $z^{Env}$, and only $z^{Inv}$ is used for label prediction, which is achieved by leveraging the empirical observation encompassing the available labeled samples.

### 6.1 CROSS-WORLD COUNTERFACTUAL CAUSALITY MODELING VIA TELEPORTER THEORY

The obtained invariant representation unavoidably contains significant environment-dependent information due to the inherent *inductive bias* arisen from the learning paradigm of benchmark methods. On the contrary, the desired invariant representation is expected to solely contain pure environment-agnostic predictive information. However, such a representation can barely be acquired in the real world yet feasibly obtained in the counterfactual world.

To this end, we propose to explore the causal mechanism behind both factual and counterfactual variables, which is accomplished by modeling the cross-world counterfactual causality. Concretely, we establish an SCM at first, as depicted in Fig. 5(a), which elaborates on the causality among the variables in GraphOOD in the real world. In Fig. 5(a), there exist four endogenous variables in the real world: the input graph $X$, the learned representation $R$ of $X$, the predicted label $Y$ and the environment-dependent information $E$. $U_X$, $U_R$, $U_E$ and $U_Y$ are four exogenous variables corresponding to the endogenous variables. According to Theorem 1, the variable $E$ can be determined as the teleporter, so the cross-world SCM is demonstrated in Fig. 5(b), where $x$ represents the intrinsic causal subgraph, $R_x$ denotes the environment-agnostic invariant representation, and $Y_x$ denotes the predicted label corresponding to the graph $x$, which is also the *true* label, since ideally, $x$ and $R_x$ only include environment-agnostic task-dependent information in the counterfactual world.

### 6.2 COUNTERFACTUAL CONDITIONAL PROBABILITY ESTIMATION VIA MULTI-SCALE MIXUP SCHEME

According to the cross-world SCM in Fig. 5(b), we determine that our objective is to derive the *ideal* predicted label $Y_x$ by only using the *available* $X$. Such an objective can be formalized as follows: computing the counterfactual probability $P(Y_x = y \mid X = x')$, where $x'$ denotes the available value of $X$, and $y$ denotes the true label of $x'$. Deriving $P(Y_x = y \mid X = x')$ is equivalent to calculating the conditional probability of $X$ on $Y_x$ in the cross-world SCM. Such a computation can

Table 1: Evaluation performance on GOOD (Gui et al., 2022) and DrugOOD (Ji et al., 2022) benchmark. The best is marked with **boldface** and the second best is with underline. † denotes the reproduction results.

| Method | GOOD-HIV | | DrugOOD | | | | Average |
| | scaffold-covariate | size-covariate | IC50-assay | IC50-scaffold | EC50-assay | EC50-scaffold | |
|---|---|---|---|---|---|---|---|
| DIR | 68.44(2.51) | 57.67(3.75) | 69.84(1.41) | 66.33(0.65) | 65.81(2.93) | 63.76(3.22) | 65.31 |
| GSAT | 70.07(1.76) | 60.73(2.39) | 70.59(0.43) | 66.45(0.50) | 73.82(2.62) | 64.25(0.63) | 67.65 |
| GREA | 71.98(2.87) | 60.11(1.07) | 70.23(1.17) | 67.02(0.28) | 74.17(1.47) | 64.50(0.78) | 68.00 |
| CAL | 69.12(1.10) | 59.34(2.14) | 70.09(1.03) | 65.90(1.04) | 74.54(4.18) | 65.19(0.87) | 67.36 |
| DisC | 58.85(7.26) | 49.33(3.84) | 61.40(2.56) | 62.70(2.11) | 63.71(5.56) | 60.57(2.27) | 59.42 |
| MoleOOD | 69.39(3.43) | 58.63(1.78) | 71.62(0.52) | 68.58(1.14) | 72.69(1.46) | 65.74(1.47) | 67.78 |
| CIGA | 69.40(1.97) | 61.81(1.68) | **71.86(1.37)** | **69.14(0.70)** | 69.15(5.79) | 67.32(1.35) | 68.11 |
| iMoLD † | 73.54(1.33) | 65.87(1.98) | 71.23(0.14) | 67.30(0.35) | 76.03(1.66) | 66.41(1.88) | 70.06 |
| iMoLD+MsMs | **74.43(1.96)** | **66.19(2.32)** | 71.70(0.62) | 67.77(0.48) | **77.29(0.65)** | **67.79(0.84)** | **70.86** |

be approximated by using neural network-based methods. Adhering Theorem 2's $d$-separation for cross-world counterfactuals, we can directly obtain $X \perp\!\!\!\perp Y_x \mid E$ from the cross-world SCM in Fig. 5(b). The calculation of $P(Y_x = y \mid X = x')$ can be acquired as follows:

$$P(Y_x = y \mid X = x') = \sum_e P(Y_x = y \mid X = x, E = e)P(E = e \mid X = x') \tag{6}$$

As demonstrated in Fig. 6, to acquire $P(Y_x = y \mid X = x')$, we propose to design a fine-grained method, which can derive the invariant part $z^{Inv}$ and the environment-dependent part $Z^{Env}$ from the input graph $x'$, thereby predict the true label $y$ by leveraging $Z^{Inv}$. According to Equation 6, $P(Y_x = y \mid X = x')$ can be estimated by summing the conditional probability of $P(Y_x = y \mid X = x, E = e)$ with respect to different environment-dependent information $E$, i.e., $z^{Env}$. Following (Zhuang et al., 2024), we utilize a contrastive learning module to estimate $P(Y_x = y \mid X = x, E = e)$, where $z^{Inv}$ is firstly concatenated by a shuffled batch of $z^{Env}$, and then projected into $\tilde{z}^{Inv}$ via a MLP-predictor $\rho$. Ultimately, $z^{Inv}$ and $\tilde{z}^{Inv}$ are used to measure the similarity for contrasting. Accordingly, expanding the available value set of $E$ can widely obtain a more precise estimation of $P(Y_x = y \mid X = x')$. Hence, we introduce a Multi-Scale Mixup Scheme (MsMs) to enrich the available data of $E$, which is achieved by leveraging a hyperparameter $scale$ in concatenating the shuffled environment $z^{Env}$. Furthermore, we further expand the available value set of $E$ by the scaled mixup scheme by $M$ times.

## 6.3 EXPERIMENTS ON GRAPHOOD

The detailed descriptions of the benchmarks and the baselines are in **Appendix** H.1 and **Appendix** H.2, respectively. To ensure reproducibility, the intricate details of our method's architecture, and our hyper-parameter settings are detailed in the **Appendix** H.3. The empirical results on the GOOD and DrugOOD benchmarks are presented in Table 1. By enhancing the available value set $E$ with MsMs, our method places the best in four of six datasets, and shows the best average ROC-AUC score among the baselines, which indicates the effectiveness of our proposed method and further emphasizes the practical generalization of teleporter theory. Testing the teleporter theory across a broader range of datasets and scenarios would further substantiate its generalizability and effectiveness. For instance, we have extended its practical application to the image domain, as detailed in the **Appendix** H.5.

## 7 CONCLUSIONS AND LIMITATIONS

We strive to explore graphical representation of counterfactuals and propose the teleporter theory to address the challenge of simultaneously representing factual and counterfactual variables in a single SCM. The cross-world SCM constructed by using the teleporter nodes can well avoid the theoretical limitations of current approaches in various cross-world counterfactual scenarios, thereby demonstrating the completeness and generalization of the teleporter theory. However, the rules that teleporter variables are required to adhere are quite stringent, and introducing such constraints increases the complexity of constructing cross-world SCMs. In future work, we will explore to simplify the proposed rules for determining teleporter variables and attempt to apply our theory in the scenarios involving multiple counterfactual worlds.

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

# A  PRELIMINARY

We recap the necessary preliminaries of causal background knowledge relevant to our work. For a more in-depth understanding, please refer to the literature (Pearl, 2009a;b; Pearl et al., 2016).

**Structural Causal Models.**  A SCM (Pearl, 2009b; Peters et al., 2017) is a causal model in a triple form, i.e., $M = \langle X, U, F \rangle$, where $U$ presents the *exogenous variable set*, determined by external factors of the model. $X = \{X_1, X_2, ..., X_n\}$ presents the *endogenous variable set*, determined by the internal functions $F = \{f_1, f_2, ..., f_n\}$. Each $f_i$ represents $\{f_i : U_i \cup PA_i \to X_i\}$, where $U_i \subseteq U$, $PA_i \subseteq X \backslash X_i$, satisfying:

$$x_i = f_i(pa_i, u_i), \quad i = 1, 2, ..., n. \tag{7}$$

$PA_i$ denotes the parent nodes of $X_i$. Note that, in SCM, *uppercase letters* conventionally denote *variables*, and *lowercase letters* conventionally denote *values* of the corresponding variables, e.g., $x_i$ is the value of $X_i$. For ease of discussion, we omit such clarification in the following sections. Each causal model $M$ corresponds to a directed acyclic graph $\mathcal{G}$, where each node corresponds to a variable in $X \cup U$, and directed edges point from $U_i \cup PA_i$ to $X_i$. It is worth noting that exogenous variables $U$ have no ancestor nodes, and each endogenous variable $X_i$ is at least a descendant of one exogenous variable.

Once we define the probability distribution of exogenous variables $U$, we can obtain the *probabilistic causal model*. A *causal world* is a tuple $\langle M, u \rangle$ where $u$ is a realization of the exogenous variables $U$, and a *probabilistic causal model* $\langle M, P(u) \rangle$ is a distribution over causal worlds.

**Interventions and Do-operator.**  The causal model $M$ describes intrinsic causal mechanisms, characterized by the observed distribution $P_M(X) = \prod_{i=1}^{n} P(x_i \mid pa_i)$. Intervention[3] is defined as forcing a variable $X_i$ to take on a fixed value $x$, modifying the model $M = \langle X, U, F \rangle$ to $M_x = \langle X, U, F_x \rangle$, where $F_x = \{F \backslash f_i\} \cup \{X_i = x\}$. This is equivalent to removing $X_i$ from its original functional mechanism $x_i = f_i(pa_i, u_i)$ and modifying this function to a constant function $X_i = x$. Formally, we denote the *intervention* as $do(x_i = x)$, called the *do-operator*. It explores how causal mechanisms will change when external interventions, or experiments, are introduced. We denote the distribution after the intervention as $P_{M_x}(X) = P(x_1, ..., x_n \mid do(x_i = x))$, where

$$P(x_1, ..., x_n \mid do(x_i = x)) = \begin{cases} \prod_{j \neq i} P(x_j \mid pa_j) & x_i = x \\ 0 & x_i \neq x \end{cases}. \tag{8}$$

**Counterfactuals.**  If $M_x$ defines the effect of the action $do(X = x)$ on $M$, what is the potential change of another endogenous variable $Y$ due to the intervention effect $M_x$? We denote $M_x$ as the SCM of the *counterfactual world* (Pearl, 2009b) derived by adopting the intervention $X = x$. The potential value of $Y$ influenced by the intervention $do(X = x)$ is denoted as $Y_x(u)$, which is a solution to the equation set $F_x$, i.e., $Y_x(u) = Y_{M_x}(u)$. Concretely, $Y_x(u)$ presents the counterfactual statement "*Under condition $u$, if $X$ were $x$, then $Y$ would be $Y_x(u)$.*"

**Path and $d$-separation.**  We recap two classic definitions (Pearl et al., 2016) to help us determine the independence between variables in the SCM graph.

**Definition 2.** *(Path) In the SCM graph, the paths from variable $X$ to $Y$ include three types of structures: 1) chain structure: $A \to B \to C$ or $A \leftarrow B \leftarrow C$; 2) fork structure: $A \leftarrow B \to C$; 3) collider structure: $A \to B \leftarrow C$.*

**Definition 3.** *($d$-separation) A path $p$ is blocked by a set of nodes $Z$ if and only if:*

1. *$p$ contains a chain of nodes $A \to B \to C$ or a fork $A \leftarrow B \to C$ such that the middle node $B$ is in $Z$, i.e., $A$ and $C$ are independent conditional on $B$, or*

2. *$p$ contains a collider $A \to B \leftarrow C$ such that the collider node $B$ is not in $Z$, and no descendant of $B$ is in $Z$, i.e., $A$ and $C$ are marginal independent.*

If $Z$ blocks every path between two nodes $X$ and $Y$, then $X$ and $Y$ are $d$-*separated*, conditional on $Z$, i.e., $X$ and $Y$ are independent conditional on $Z$, denoted as $X \perp\!\!\!\perp Y \mid Z$.

---

[3] The definition here refers to the atomic intervention (Pearl, 2009b). For brevity, we intervene on only one variable.

# B  QUANTITATIVE ANALYSIS WITH NUMERICAL EXAMPLES FOR INAPPLICABILITY OF TWIN NETWORK

## B.1  ORIGINAL FIRING SQUAD EXAMPLE IN "CAUSALITY"

Considering the firing squad example in Fig. 1(a), $A$ and $B$ are the officers, $C$ is the captain (waiting for the court order $U$), and $D$ represents the condemned prisoner. The exogenous variables are only $U$ and $W$, which represent the court order and the nervousness of police officer $A$, respectively. The values and meanings of each variable are as follows:

1. $A(u,w), B(u,w)$ indicate whether officers $A$ and $B$ fire their guns, respectively, and $D(u,w) = 1$ indicates the death of the prisoner. The prisoner will not die from any other factors besides the executioners, so we ignore the exogenous variables for $D$.

2. $D_0(u,w)$ and $D_1(u,w)$ represent the counterfactual values under interventions $A = 0$ and $A = 1$, respectively.

3. $P(u = 1) = p$ represents the probability of issuing a death sentence, $P(w = 1) = q$ represents the probability that officer $A$ pulls the trigger due to nervousness. For the specific values of the variables, please refer to Table 2.

| $u$ | $w$ | $A(u,w)$ | $D(u,w)$ | $B(u,w)$ | $D_0(u,w)$ | $D_1(u,w)$ |
|---|---|---|---|---|---|---|
| 0 | 0 | 0 | 0 | 0 | 0 | 1 |
| 0 | 1 | 1 | 1 | 0 | 0 | 1 |
| 1 | 0 | 1 | 1 | 1 | 1 | 1 |
| 1 | 1 | 1 | 1 | 1 | 1 | 1 |

Table 2: Numerical examples demonstrate the inapplicability of the twin network, with the SCM graph shown in (Pearl, 2009b), p. 213, figure 7.2 and Fig. 1. $P(u = 1) = p$ represents the probability of issuing a death sentence, and $P(w = 1) = q$ represents the probability that officer $A$ pulls the trigger due to nervousness. $A(u,w) = 1$ and $B(u,w) = 1$ indicate that officers $A$ and $B$ fire their guns, respectively. $D(u,w) = 1$ indicates the death of the prisoner, and $D_0(u,w)$ and $D_1(u,w)$ represent the counterfactual values under interventions $A = 0$ and $A = 1$, respectively. It can be verified that $P(D_a|B) = P(D_a|B, A)$, which implies $\mathbf{A} \perp\!\!\!\perp \mathbf{D_a}|\mathbf{B}$.

In this model, the distribution of the exogenous variables is

$$
P(u,w) = \begin{cases} pq, & u = 1, w = 1 \\ p(1-q), & u = 1, w = 0 \\ (1-p)q, & u = 0, w = 1 \\ (1-p)(1-q), & u = 0, w = 0. \end{cases} \tag{9}
$$

Verify that $\mathbf{D_a} \not\perp\!\!\!\perp \mathbf{A}$:

$$
\begin{aligned}
P(D_0 = 1) &= \sum_{\{(u,w)|D_0(u,w)=1\}} P(u,w) \\
&= P(u = 1, w = 0) + P(u = 1, w = 1) \\
&= p(1-q) + pq = p
\end{aligned}
$$

$$
\begin{aligned}
P(A = 1) &= \sum_{\{(u,w)|A(u,w)=1\}} P(u,w) \\
&= 1 - P(u = 0, w = 0) \\
&= 1 - (1-p)(1-q)
\end{aligned}
$$

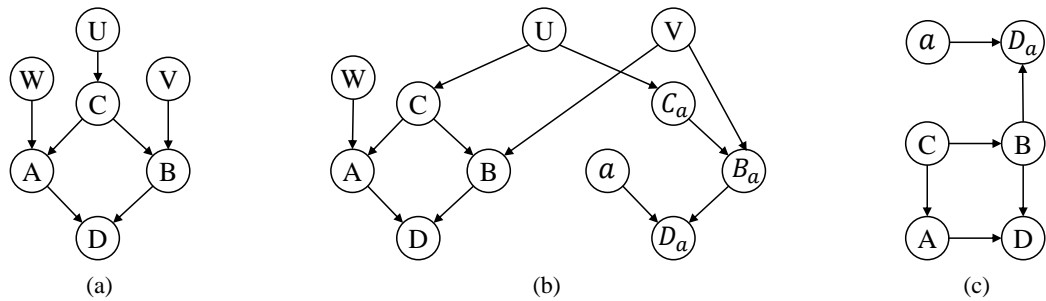

Figure 7: The SCM graph of the extended firing squad example: Figure (a) represents the real-world SCM considering all relevant exogenous variables, Figure (b) shows the cross-world SCM constructed using the twin network with all exogenous variables, and Figure (c) illustrates the cross-world SCM constructed using the teleporter theory.

$$P(D_0 = 1, A = 1) = \sum_{\{(u,w)|D_0(u,w)=1\&A(u,w)=1\}} P(u,w)$$
$$= P(u = 1, w = 0) + P(u = 1, w = 1)$$
$$= p(1 - q) + pq = p$$

Hence,

$$P(D_0, A = 1) = p \neq p(1 - (1 - p)(1 - q)) = P(D_0 = 1)P(A = 1).$$

Verify that $\mathbf{A} \perp\!\!\!\perp \mathbf{D_a}|\mathbf{B}$:

$$P(D_0 = 1|B = 1) = 1, P(D_0 = 1|B = 1, A = 1) = 1,$$
$$P(D_0 = 0|B = 1) = 0, P(D_0 = 0|B = 1, A = 1) = 0.$$

The remaining values can be verified, so $P(D_a|B) = P(D_a|B, A)$.

## B.2 EXTENDED FIRING SQUAD EXAMPLE: CONSIDERING ALL EXOGENOUS VARIABLES

The original definition of the twin network includes all exogenous variables, and clearly, the firing squad example can be further extended to better reflect real-world scenarios. Based on Fig. 1, we now consider that officer $B$ may also fire due to nervousness, and thus we introduce the exogenous variable $V$ to represent officer $B$'s nervousness. $P(v = 1) = s$ represents the probability that officer $B$ pulls the trigger due to nervousness. We still do not consider exogenous variables for $D$, as the prisoner's death is unrelated to external factors, such as the unlikely possibility of dying suddenly from illness or fear. Thus, the extended firing squad example with all exogenous variables is shown in Fig. 7. For the specific values of each variable, please refer to Table 3.

In the extended firing squad example, we rewrite the distribution of exogenous variables from equation 9:

$$P(u,w,v) = \begin{cases} pqs, & u = 1, w = 1, v = 1 \\ p(1-q)s, & u = 1, w = 0, v = 1 \\ pq(1-s), & u = 1, w = 1, v = 0 \\ p(1-q)(1-s), & u = 1, w = 0, v = 0 \\ (1-p)qs, & u = 0, w = 1, v = 1 \\ (1-p)(1-q)s, & u = 0, w = 0, v = 1 \\ (1-p)q(1-s), & u = 0, w = 1, v = 0 \\ (1-p)(1-q)(1-s), & u = 0, w = 0, v = 0. \end{cases} \tag{10}$$

Verify that $\mathbf{D_a} \not\perp\!\!\!\perp \mathbf{A}$:

| $u$ | $w$ | $v$ | $A(u,w)$ | $B(u,w)$ | $D(u,w)$ | $C(u,w)$ | $D_0(u,w)$ | $D_1(u,w)$ |
|---|---|---|---|---|---|---|---|---|
| 0 | 0 | 0 | 0 | 0 | 0 | 0 | 0 | 1 |
| 0 | 1 | 0 | 1 | 0 | 1 | 0 | 0 | 1 |
| 0 | 0 | 1 | 0 | 1 | 1 | 0 | 1 | 1 |
| 0 | 1 | 1 | 1 | 1 | 1 | 0 | 1 | 1 |
| 1 | 0 | 0 | 1 | 1 | 1 | 1 | 1 | 1 |
| 1 | 1 | 0 | 1 | 1 | 1 | 1 | 1 | 1 |
| 1 | 0 | 1 | 1 | 1 | 1 | 1 | 1 | 1 |
| 1 | 1 | 1 | 1 | 1 | 1 | 1 | 1 | 1 |

Table 3: The numerical example of the extended firing squad, with its SCM graph depicted in Fig. 7, includes the exogenous variable $V$ for officer $B$. Compared to the original SCM graph depicted in Fig. 1, this model additionally considers the probability that officer $B$ fires due to nervousness, which is $P(v = 1) = s$.

$$P(D_0 = 1) = \sum_{\{(u,w,v)|D_0(u,w,v)=1\}} P(u,w,v)$$
$$= 1 - P(u = 0, w = 0, v = 0) - P(u = 0, w = 1, v = 0)$$
$$= 1 - (1 - p)(1 - s) = p + s - ps$$

$$P(A = 1) = \sum_{\{(u,w,v)|A(u,w,v)=1\}} P(u,w,v)$$
$$= 1 - P(u = 0, w = 0, v = 0) - P(u = 0, w = 0, v = 1)$$
$$= 1 - (1 - p)(1 - q) = p + q - pq$$

$$P(D_0 = 1, A = 1) = \sum_{\{(u,w,v)|D_0(u,w,v)=1 \& A(u,w,v)=1\}} P(u,w,v)$$
$$= 1 - P(u = 0, w = 0, v = 0) - P(u = 0, w = 0, v = 1) - P(u = 0, w = 1, v = 0)$$
$$= p + sq - pqs$$

Hence,

$$P(D_0, A = 1) = (p + s - ps)(p + q - pq) \neq p + sq - pqs = P(D_0 = 1)P(A = 1).$$

Verify that $\mathbf{A} \perp\!\!\!\perp \mathbf{D_a}|\mathbf{B}$:

$$P(D_0 = 1|B = 1) = 1, P(D_0 = 1|B = 1, A = 1) = 1,$$
$$P(D_0 = 0|B = 1) = 0, P(D_0 = 0|B = 1, A = 1) = 0.$$

The remaining values can be verified, so $P(D_a|B) = P(D_a|B, A)$.

## C  COMPARISON WITH ROBINS'S SINGLE WORLD INTERVENTION GRAPHS (SWIG)

We will illustrate the superiority and consistency of our work compared to SWIG through three examples. The teleporter theory not only accommodates the SWIG and Twin Network frameworks but also addresses their deficiencies, constructing a comprehensive graphical model for identifying the counterfactual conditional independence. These three examples can be found in Fig. 8–10.

1. Fig. 8: According to the SWIG model, we obtain Fig. 8 (b). When the factual variable $Z$ is given, since the SWIG model cannot fully represent all factual variables, we are unable to

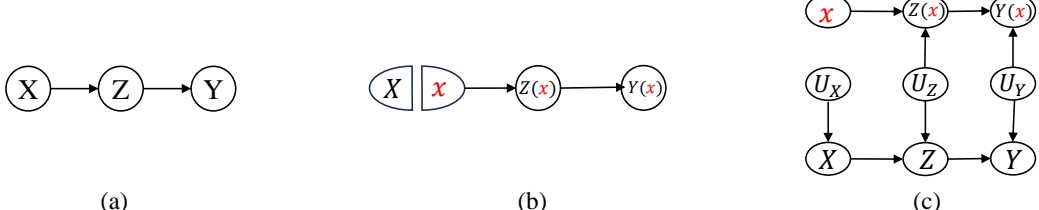

Figure 8: (a) The DAG $\mathcal{G}$, Figure 4.3 in (Pearl et al., 2016, p.99); (b) The SWIG model cannot directly derive $X \not\perp Y(x)|Z$, a conclusion drawn from (Pearl et al., 2016, p.103). This is because the factual variable $Z$ is not present in the graph; (c) Teleporter model can obtain $X \not\perp Y(x)|Z$, which is consistent with Pearl's conclusion.

directly obtain independence relationships conditioned on the factual variable $Z$ from the graphical model. However, according to (Pearl et al., 2016, p.99), $X \not\perp Y(x)|Z$, which can also be proved by introducing quantitative analysis with numerical examples. This indicates that SWIG is limited when dealing with the real-world descendants of $X$. In contrast, according to the teleporter theory as shown in Fig. 8 (c), $X \to Z \leftarrow U_Z \to Z(x) \to Y(x)$ is unblocked when given $Z$.

2. Fig. 9, or in Fig.7 on page 7 of Richardson & Robins (2013): According to the SWIG model, we can infer $X \perp\!\!\!\perp Y(x)|L_1$ and $X \perp\!\!\!\perp Y(x)|L_1, L_2(x)$, but it cannot obtain $X \not\perp Y(x)|L_1, L_2$, which can also be proved by introducing quantitative analysis with numerical examples. In other words, when all the factual variables are given, SWIG is limited. However, according to the teleporter theory, as shown in Fig. 9 (c), when $X \to Y \leftarrow U_Y$ and the collider node $Y$'s descendants $L_2$ are given, by Pearl et al. (2016) p.44 (Rule 3), $X$ and $U_Y$ are dependent, thus $X$ and $Y(x)$ are dependent, so $X \not\perp Y(x)|L_1, L_2$[4].

3. Fig. 10: According to the SWIG model, we obtain Fig. 10 (b). $Z(x_0)$ blocks the only path from $X_1(x_0)$ to $Y(x_0, x_1)$. Given the consistency conditions, we obtain $Y(x_0, x_1) \perp\!\!\!\perp X_1|Z, X_0 = x_0$. However, according to the teleporter theory, as shown in Fig. 10 (c), the path $X_1 \leftarrow H \to Z(x_0) \to Y(x_0, x_1)$ is not blocked, so $Y(x_0, x_1) \not\perp X_1|Z, X_0 = x_0$, **which aligns with the conclusion in Example 11.3.3** on p.353 of (Pearl, 2009b) and can also be proved by introducing quantitative analysis with numerical examples. Based on this example in Fig. 10, we summarize the limitations of SWIG when dealing with multiple worlds, especially the need for additional consistency assumptions and **its inability to intuitively represent all variables in a single graph.**

We acknowledge that SWIG's construction is more streamlined. Although teleporter theory follows more criteria, it can handle conditional independencies between any two **cross-world** variables. As demonstrated in Fig. 8 and 9, SWIG's limitations in handling null hypotheses are evident, such as its inability to manage conditional independencies when descendants of $X$ in the real world are given.

SWIG's ability to handle variables is limited, making it challenging to encompass **all** variables' conditional independencies in both the real and counterfactual worlds, whereas the teleporter theory offers a more generalized approach.

---

[4]This conclusion is derived from the Non-Parametric Structural Equation Models with Independent Errors (NPSEM-IE) considered in (Pearl, 2009b).

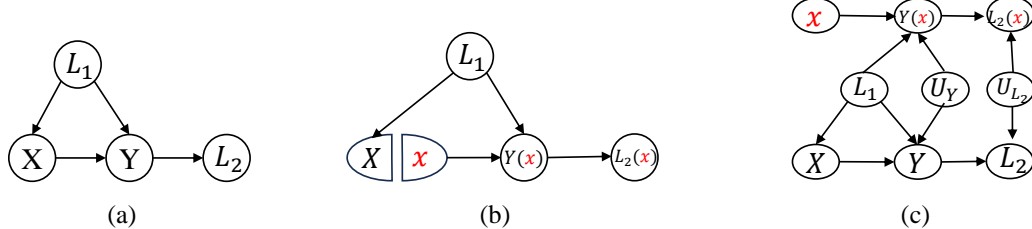

Figure 9: (a) The DAG $\mathcal{G}$, Figure 7 in (Richardson & Robins, 2013, p.7); (b) SWIG model shows that $X \perp\!\!\!\perp Y(x)|L_1$ **but does not imply** $X \not\perp\!\!\!\perp Y(x)|L_1, L_2$; (c) Teleporter model can obtain $X \not\perp\!\!\!\perp Y(x)|L_1, L_2$.

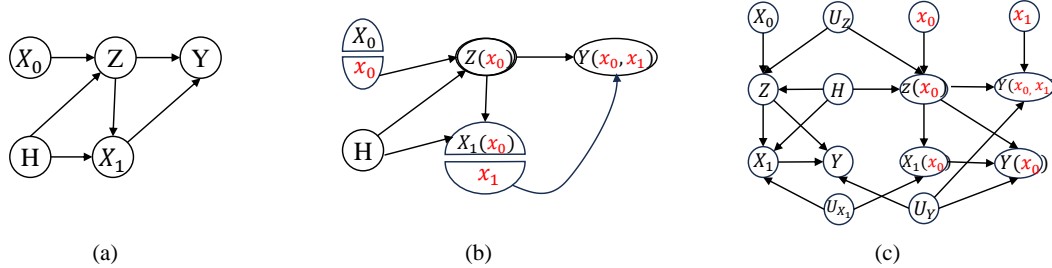

Figure 10: (a) The DAG $\mathcal{G}$, Ex.11.3.3, Fig.11.12 in (Pearl, 2009b, p.353); (b) SWIG model shows that $Y(x_0, x_1) \perp\!\!\!\perp X_1|Z, X_0 = x_0$; (c) Teleporter model obtains the **same conclusion as Pearl:** $Y(x_0, x_1) \not\perp\!\!\!\perp X_1|Z, X_0 = x_0$.

# D   DEEPER ANALYSIS ON THE THEORETICAL APPLICATIONS OF TELEPORTER THEORY.

## D.1   THE CLARIFICATION BETWEEN FACTUAL VARIABLE $D$ AND COUNTERFACTUAL VARIABLE $D_x$

The factual variable $D$ refers to the collected observed data, encompassing different groups with various values of $X$. The counterfactual variable $D_x$ corresponds to the unique group after the intervention $X = x$.

For instance, consider the numerical example corresponding to Fig. 8. $E(Y_{X=1}|Z = 1)$ represents the expected salary for individuals with a skill level $Z = 1$ if they had received higher education. In this scenarios, these individuals with $Z = 1$, there exist both the ones who have received higher education ($X = 1$) and the ones who have not ($X = 0$). In contrast, the expectation $E(Y|do(X = 1), Z = 1)$ refers to the group of individuals in the post-intervention world, which only includes the ones who have received higher education($X = 1$) (i.e., after intervening on $X = 1$, we then condition on $Z = 1$). Since $E(Y|do(X = 1), Z = 1)$ only represents the post-intervention world. $E(Y_{X=1}|Z = 1)$ represents a cross-world scenario, but do-operator cannot capture counterfactual queries: $\mathbf{E(Y|do(X = 1), Z = 1)} \neq \mathbf{E(Y_{X=1}|Z = 1)}$. $E(Y|do(X = 1), Z = 1)$ can be easily converted into the counterfactual notation $E(Y_{X=1}|Z_{X=1} = 1)$, where $\mathbf{Z_{X=1}}$ explicitly designates the event $Z = 1$ in the post-intervention world. This leads to $\mathbf{E(Y_{X=1}|Z_{X=1} = 1)} \neq \mathbf{E(Y_{X=1}|Z = 1)}$, which is why we believe it is necessary to distinguish between factual variable $Z = 1$ and counterfactual variable $\mathbf{Z_{X=1} = 1}$. Therefore, after intervention, we transform $Z$ into $Z_x$, and in the cross-world SCM graph, $Z$ and $Z_x$ are represented as two distinct nodes.

### D.2 OBTAIN CONDITIONAL EXOGENEITY TO CONTROL FOR CONFOUNDING BIAS AND IDENTIFY THE CORRECT ADJUSTMENT VARIABLES

We believe that an important use lies in more intuitively achieving conditional exchangeability (or exogeneity) to control for confounding bias, which is where the teleporter theory and SWIG are in agreement. Furthermore, the teleporter theory, by knowing the independence of all variables, helps avoid incorrect adjustments of factual variables.

We compute $P(Y_{x'}|x, y)$ or $P(Y_{x'}|y)$ in the extended firing squad example, corresponding to Fig. 7(a), which is $P(D_0 = 1|A = 1, D = 1)$. According to the teleporter theory, as shown in Fig. 7(c), upon conditioning on $D$, new paths between $A$ and $D_a$ are opened through the collider node $D$. Consequently, conditional on $\{D, C\}$, $A$ and $D_a$ are not $d$-separated, leading to $A \not\perp\!\!\!\perp D_a \mid \{D, C\}$. However, when conditioned on $\{D, B\}$, $A$ and $D_a$ become $d$-separated, satisfying $A \perp\!\!\!\perp D_a \mid \{D, B\}$. **Therefore, we can only choose $B$ to control for confounding bias, rather than $C$.** However, in the twin network, as shown in Fig. 7(b), when conditioned on $D$, there are two open paths between $A$ and $D_a$: $A \rightarrow \underline{D} \leftarrow B \leftarrow V \rightarrow B_a \rightarrow D_a$ and $A \leftarrow C \leftarrow U \rightarrow C_a \rightarrow B_a \rightarrow D_a$. In this case, arbitrarily choosing a variable from $\{B, C\}$ for adjustment is not valid because $A \not\perp\!\!\!\perp D_a \mid \{D, B\}$ and $A \not\perp\!\!\!\perp D_a \mid \{D, C\}$. This is where the twin network fails in cross-world adjustment. For verification of this conclusion, please refer to the following numerical calculations.

As shown in Fig. 7(a) and Table 3:

$$P(D_0 = 1|A = 1, D = 1) = \frac{P(D_0 = 1, A = 1, D = 1)}{P(A = 1, D = 1)}$$
$$= \frac{1 - (1-p)(1-q) - (1-p)q(1-s)}{1 - (1-p)(1-q)} \tag{11}$$

1. Choose $B$ to control for confounding bias:
$$P(D_0 = 1|A = 1, D = 1) = P(D = 1|A = 0, D = 1, B = 1)P(B = 1|A = 1, D = 1)$$
$$+ P(D = 1|A = 0, D = 1, B = 0)P(B = 0|A = 1, D = 1)$$
$$= \frac{1 - (1-p)(1-q) - (1-p)q(1-s)}{1 - (1-p)(1-q)} \tag{12}$$

2. Choose $C$ to control for confounding bias:
$$P(D_0 = 1|A = 1, D = 1) = P(D = 1|A = 0, D = 1, C = 1)P(C = 1|A = 1, D = 1)$$
$$+ P(D = 1|A = 0, D = 1, C = 0)P(C = 0|A = 1, D = 1)$$
$$= \frac{(1-p)q}{1 - (1-p)(1-q)} \tag{13}$$

It is evident that using $C$ as adjustment to calculate $P(D_0 = 1|A = 1, D = 1)$ is incorrect, which aligns with the conclusions drawn from our teleporter theory.

### D.3 COMPUTATION OF COUNTERFACTUAL QUERIES

Another significant potential of Teleporter theory is in computing complex counterfactual queries. The standard approach, which encompasses Abduction, Action, and Prediction, albeit correct, is computationally expensive.

Typically, when we aim to compute $P(Y_x = y|E = e)$, we need to obtain the distribution of exogenous variables, i.e., $\langle M_x, P(u|e) \rangle$. However, teleporter theory allows us to construct a cross-world network, reducing the problem to computing a conditional probability $P(y^*|e)$ in an augmented Bayesian network. This computation can be performed using standard evidence propagation techniques, leveraging conditional independence and adopting a local computation approach.

We have identified a neural network architecture constrained by twin network: deep twin network (Vlontzos et al., 2023), which is a neural network implementation of the aforementioned Bayesian inference techniques. Teleporter theory removes most exogenous variables while preserving the topology of the cross-world network, which can significantly reduce the graph size required for inference in the twin network for counterfactual queries. This demonstrates that our teleporter theory also has the potential to be combined with neural networks for estimating counterfactuals.

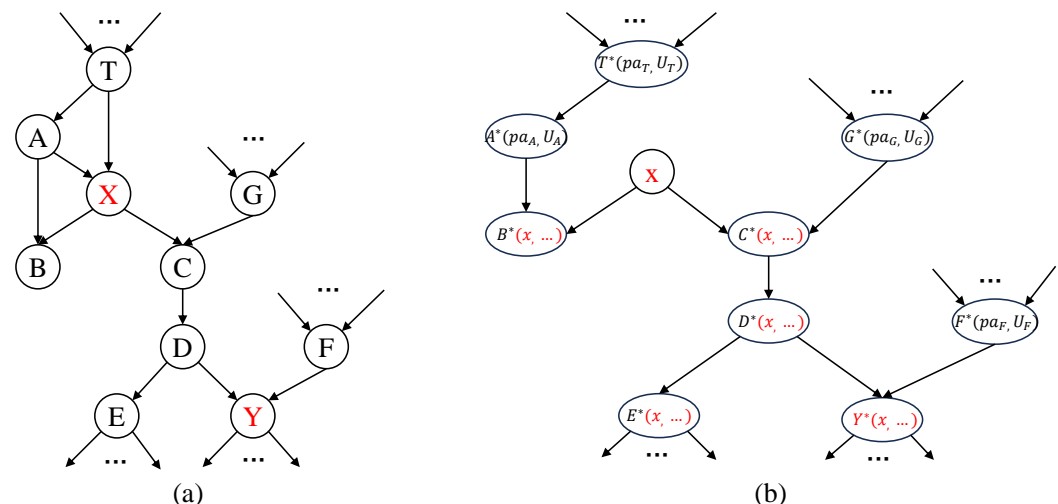

Figure 11: (a) The SCM graph of the real world $\mathcal{W}_r$; (b) The illustration for the two types of variables in the counterfactual world $\mathcal{W}_c$, where $x$ is highlighted in red to indicate that the value of the structural equation is determined by $X = x$.

# E  PROOF OF THEOREM 1

The core idea of the proof is to find pairs of variables that have the **same values** in both the real world and the counterfactual world: $Z \leftarrow U_Z \rightarrow Z^*$ (note that their structural equations are certainly the same, as the only difference in structural equations between the two worlds is the intervened variable). We implement a **merging operation** on these pairs of variables (in the cross-world SCM graph, this is represented by merging three variables into one variable $Z$), thus $Z$ is called the Teleporter.

*Proof of Lemma 1.* To visually illustrate the types of variables in the real world $\mathcal{W}_r$ and the counterfactual world $\mathcal{W}_c$, refer to the example in Fig. 11. In the real world $\mathcal{W}_r$, as shown in Fig. 11(a), we categorized the relationship of specific variables $X$ with the rest of the variables in the SCM graph into four types: descendant nodes, d-separated nodes, sibling nodes, and parent nodes. Since $X$ is intervened, the arrow pointing to $X$ is removed in the counterfactual world $\mathcal{W}_c$, as shown in Fig. 11(b). Therefore, variables related to $x$ are either its descendants or $d$-separated from it, while parent and sibling nodes naturally transform into $d$-separated from $x$. □

*Proof of Theorem 1.* We first use proof by contradiction to demonstrate the cross-world invariance property of the teleporter $Z$. Since there are only two types of variables in the counterfactual world $\mathcal{W}_r$: $Z^*$ and $D^*$,

- The set of descendants of the intervention variable $X = x$, denoted as $D^*$. For the variables in the set $D^*$, their values are given by $d^* = f^x_{D^*}(pa_{D^*}, u_{D^*}) = f_{D^*}(f^x_{D^*_{j_1}}(pa_{j_1}, u_{j_1}), f^x_{D^*_{j_2}}(pa_{j_2}, u_{j_2}), \ldots, f^x_{D^*_{j_n}}(pa_{j_n}, u_{j_n}), u_{D^*})$. There must exist at least one parent node $D^*_{j_k}$ of $D^*$ whose structural equation value is determined by $X = x$, i.e., $D^*_{j_k} = f^x_{D^*_{j_k}}(x, \ldots, u_{j_k})$. This can be obtained by iteratively applying the structural equations until ultimately recursing to $X = x$. Hence, we denote $D^*$ as $D_x$ to indicate that its values differ from those in the real world $\mathcal{W}_r$.

- The set of variables $d$-separated from the intervention variable $X = x$, denoted as $Z^*$. Similar to the structural equations of $D^*$, we need to prove that the values of the structural equations for **any** parent node $Z^*_{i_k}$ of $Z^*$ are not influenced by $X = x$. Using a proof by contradiction, assume there exists a $Z^*_{i_k} = f^x_{Z^*_{i_k}}(x, \ldots, u_{i_k})$. Then $X = x$ is $d$-connected to $Z^*_{i_k}$, and since $Z^*_{i_k}$ is a parent node of $Z^*$, $X = x$ would be $d$-connected to $Z^*$, which contradicts the definition of $Z^*$. Therefore, the values of the variable $Z^*$ in the counterfactual world are equal to its values in the real world, i.e., $Z = Z^*$.

Since in the real world $\mathcal{W}_r$, the structural equation and value of the teleporter $Z$ are equal to the corresponding structural equation and value of $Z^*$ in the counterfactual world $\mathcal{W}_c$, the exogenous variable $U_Z$ is no longer needed as a unique proxy for the counterfactual variable $Z^*$. Instead, $Z^*$ is governed by the equation $Z = f_Z$. Therefore, in the cross-world SCM graph, $Z \leftarrow U_Z \rightarrow Z^*$ merges into a single node $Z$. The topological structure of the cross-world SCM graph $\mathcal{W}_m$ is still determined by $\mathcal{W}_r$ and $\mathcal{W}_c$, preserving the connectivity between variables. □

## F  PROOF OF THEOREM 2

After constructing the DAG $\mathcal{G}_m$, which consists of both factual and counterfactual variables using the teleporter theory, the resulting cross-world Bayesian network preserves the topological structure of both the real-world $\mathcal{W}_r$ and the counterfactual world $\mathcal{W}_c$ (i.e., the directed relationships between variables in the DAG). Therefore, the probability function $P_m$ and the DAG $\mathcal{G}_m$ are Markov compatible (Pearl, 2009b, Def.1.2.2). As a result, the $d$-separation criterion naturally extends to the cross-world SCM graph. □

## G  PROOF OF THEOREM 3

Below, we use the calculation of the counterfactual statement $P(Y_x = y \mid E = e)$ as an example to illustrate that once the conditional independence of relevant variables is obtained through cross-world SCM, cross-world adjustment can be achieved using simple algebraic derivations:

$$P(Y_x = y \mid E = e) = \sum_z P(Y_x = y \mid E = e, Z = z)P(Z = z \mid E = e) \tag{14}$$

$$= \sum_z P(Y_x = y \mid X = x, E = e, Z = z)P(Z = z \mid E = e) \tag{15}$$

$$= \sum_z P(Y = y \mid X = x, E = e, Z = z)P(Z = z \mid E = e). \tag{16}$$

Equation 15 holds because $X \perp\!\!\!\perp Y_x \mid \{E, Z\}$. Equation 16 holds due to the consistency condition: $X(u) = x, Y(u) = y \rightarrow Y_x(u) = y$. □

## H  EXPERIMENTAL SETTINGS

### H.1  BENCHMARKS

We employ two real-world GraphOOD benchmarks, i.e. GOOD (Gui et al., 2022) and DrugOOD (Ji et al., 2022) to exam the performance of our method. GOOD is a systematic benchmark which is tailored specifically for graph OOD problems. We adopt one molecular dataset GOOD-HIV for the graph prediction task, where the objective is binary classification to predict whether a molecule can inhibit HIV. DrugOOD is an OOD benchmark for AI-aided drug discovery, which provides two measurements (IC50 and EC50) and their environment-splitting strategies (assay, scaffold, and size). According to the split strategy, we choose four datasets, e.g., IC50-assay, IC50-scaffold, EC50-assay, EC50-scaffold as the benchmarks. Due to the chosen task of GOOD-HIV and DrugOOD are both binary classification, we adopt the ROC-AUC score as the evaluation metric. The details of benchmark are shown in Table 4.

Table 4: Benchmark statistics. BC denotes Binary Classification.

| Dataset | | | Task | Metric | #Train | #Val | #Test | #Tasks |
|---|---|---|---|---|---|---|---|---|
| GOOD | HIV | scaffold-covariate | BC | ROC-AUC | 24682 | 4133 | 4108 | 1 |
| | | size-covariate | BC | ROC-AUC | 26169 | 4112 | 3961 | 1 |
| DrugOOD | IC50 | assay | BC | ROC-AUC | 34953 | 19475 | 19463 | 1 |
| | | scaffold | BC | ROC-AUC | 22025 | 19478 | 19480 | 1 |
| | EC50 | assay | BC | ROC-AUC | 4978 | 2761 | 2725 | 1 |
| | | scaffold | BC | ROC-AUC | 2743 | 2723 | 2762 | 1 |

## H.2 BASELINES

To compare our method with other methods, we include three interpretable graph learning methods (DIR (Wu et al., 2022), GSAT (Miao et al., 2022) and GREA (Liu et al., 2022b)) and five GraphOOD algorithms (CAL (Sui et al., 2022), DisC (Fan et al., 2022), MoleOOD (Yang et al., 2022), CIGA (Chen et al., 2022) and iMoLD (Zhuang et al., 2024)) as baselines. Note that iMoLD performs not stable on DrugOOD benchmarks, so we reproduce the results using the official code on github. The descriptions and the github links of the baselines are listed as follows:

- **DIR** (Wu et al., 2022) identifies an invariant rationale by performing interventional data augmentation to generate multiple distributions from the subset of a graph. `https://github.com/Wuyxin/DIR-GNN`

- **GSAT** (Miao et al., 2022) introduces an interpretable graph learning method that leverages the attention mechanism. It injects stochasticity into the attention process to select subgraphs relevant to the target labels. `https://github.com/Graph-COM/GSAT`

- **GREA** (Liu et al., 2022b) identifies subgraph structures called rationales by employing an environment replacement technique. This allows the generation of virtual data points, which in turn enhances the model's generalizability and interpretability. `https://github.com/liugangcode/GREA`

- **CAL** (Sui et al., 2022) introduces a causal attention learning strategy for graph classification tasks. This approach encourages GNNs to focus on causal features, while mitigating the impact of shortcut paths. `https://github.com/yongduosui/CAL`

- **DisC** (Fan et al., 2022) takes a causal perspective to analyze the generalization problem of GNNs. It proposes a disentangling framework that learns to separate causal substructures from biased substructures within graph data. `https://github.com/googlebaba/DisC`

- **MoleOOD** (Yang et al., 2022) investigates the OOD problem in the domain of molecules. It designs an environment inference model and a substructure attention model to learn environment-invariant molecular substructures. `https://github.com/yangnianzu0515/MoleOOD`

- **CIGA** (Chen et al., 2022) proposes an information-theoretic objective that extracts the desired invariant subgraphs from the causal perspective. `https://github.com/LFhase/CIGA`

- **iMoLD** (Zhuang et al., 2024) propose a first-encoding-then-split method to disentangle the invariant representation and the environment representation via a residual vector quantization skill and a self-supervised learning pattern. `https://github.com/HICAI-ZJU/iMoLD`

| Algorithm | CMNIST | VLCS | PACS | OfficeHome | Average |
|---|---|---|---|---|---|
| ERM | 51.5 ± 0.1 | 77.5 ± 0.4 | 85.5 ± 0.2 | 66.5 ± 0.3 | 70.25 |
| balance+ERM | 60.1 ± 1.0 | 76.1 ± 0.3 | 85.2 ± 0.4 | 67.1 ± 0.4 | 72.13 |
| balance+ERM+ours | **62.5 ± 2.5** | **77.1 ± 2.2** | **86.3 ± 1.2** | **68.2 ± 0.8** | **73.53** |

Table 5: Performance on image domain.

## H.3 HYPER-PARAMETERS

We reproduce iMoLD with the best hyper-parameters provided in the paper. As for the MsMs part, we choose $scale$ from {0.3, 0.7, 1.0}, $M$ from {1, 3, 5}.

## H.4 EXPERIMENTS COMPUTE RESOURCES

Experiments are conducted on one 24GB NVIDIA RTX 4090 GPU.

## H.5 TEST THE TELEPORTER THEORY ON A WIDER RANGE OF DATASETS BEYOND GOOD AND DRUGOOD

We expand the pactical application into the image domain. In image classification tasks, we typically hope that the neural network will focus on the semantic parts (foreground) of an image and ignore

the background information. This is because background information is generally easier to learn, and when the foreground and background of a class of images frequently appear together, the neural network will be more inclined to learn the background information, which can lead to poorer performance on image OOD (domain generalization) tasks. Based on this characteristic, the image OOD field can also construct an SCM model as Fig. 5(a), where $X$ represents the input image, $E$ represents the background information, $R$ represents the semantic information, and $Y$ represents the predicted label. Similar to the analysis of the graph domain, here the teleporter theory can be used to analyze that the background $E$ is a transworldly backdoor variable. According to Equation 6, we can expand the range of $E$ to obtain a more accurate causal effect of $P(Y_x = y | X = x')$.

Specifically, we firstly pass the input image through a fast Fourier transform to separate the foreground and background, and then swap the foreground and background of different images within a batch to expand the values of $E$. We use balance+ERM from (Wang et al., 2022) as the baseline, and verify the effectiveness of our method on four benchmark domain generalization datasets on OOD scenarios: CMNIST (Arjovsky et al., 2019), VLCS (Fang et al., 2013), PACS (Li et al., 2017) and OfficeHome (Venkateswara et al., 2017), with the results shown in the Table 5 (repeat for 3 times). This experiment demonstrates that our method is not only effective in the graph domain, but can also be applied to other domains, further illustrating the generality of our method.

