# OpenReview forum: "Teleporter Theory: A General and Simple Approach for Modeling Cross-World Counterfactual Causality"
_ICLR.cc/2025/Conference — ICLR 2025 Conference Withdrawn Submission_

### Official Review · Reviewer_HYtH · 2024-11-01

**Soundness:** 2
**Presentation:** 3
**Contribution:** 4
**Rating:** 6
**Confidence:** 3

**Summary:**

In their work, the authors demonstrate the inability of twin network and SWIGs the truthfully represent conditional independencies with regard to some underlying counterfactual distribution. In the following, the authors propose a 'teleporter theory', a novel graphical representation that is able to alleviate these problems and allows for generalized adjustment within cross-world inference.

The authors define two categories of variables that change ($D\neq D^*$) or remain constant ($Z=Z^*$) between the factual and counterfactual world. As variables $Z, Z^*$ adapt the same values in the observed and counterfactual domain, the authors propose to merge both sets into unique 'teleporter' variables that are then shared between both worlds. The resulting 'teleporting networks' are claimed to be faithful with regard to the underlying counterfactual distribution. Due to the obtained faithfulness, a generalized cross-world adjustment between any two variables is defined. As a result, the authors demonstrate that broader sets of variables can be used for counterfactual cross-world adjustment than what was available with previous twin networks.

The authors propose to train neural based predictors via a contrastive learning approach that leverages their newly presented adjustment method. The models are tested to two graph OOD datasets and for simple image scenarios. Results are compared against several baselines.

**Strengths:**

The paper contains important theoretical contributions, by identifying and fixing issues of Markov compatibility of previous approaches. The presented improvements open up potentials for better utilization of data across many practical applications. The presented experimental evaluation seems to underline the theoretical claims. The presented teleporter networks --in particular with theorem 3-- allow for generalized cross-world adjustments between any two variables and therefore generalize upon backdoor adjustments within twin networks.

The identified issues regarding representation of independencies within twin network and (while I'm not an expert in) SWIGs, are inferred clearly from examples. The presented theorems on teleporter theory and cross-world adjustment are stated clearly and, to the best of my knowledge, are sound.

Apart from the clear theoretical benefits, the authors seem to show that their approach can perform consistently above other baseline methods for graph OOD experiments and simple image datasets.

**Weaknesses:**

While the paper contains intriguing novel theoretical results, some minor details on the presented theory are left open. My main concern is with the description of the experimental evaluation:

1) The identified limitations in section 3.1 imply that $(X \text{ indep } Y | Z)_p \implies (X \text{ indep } Y | Z)_G$ may not always hold for twin networks. To the best of my knowledge, this condition is usually referred to as faithfulness of the graph. While the proof of theorem 2 mentions Markov compatibility, a discussion and explicit mention of faithfulness (or differences to it) could improve the section.

2) The requirement of rule 3 in theorem 1 is not obvious to me. Removing the previously shared $U_Z$, but fixing $Z$ instead, on one hand it simplifies the graph, on the other hand possible information about $U_Z$ is lost. To the best of my understanding, there would be no downside in the following proofs of keep $U_Z$ and having rule 3 as an optional step.

3) Description of model training is held very short. The authors propose a Multi-Scale Mixup Scheme (MsMs) on top of an existing algorithm that leverages their cross-world adjustment. While the general idea is well introduced, details and the exact working of the MsMs are lacking.

**Questions:**

My questions generally concern the weaknesses stated above. In particular, I would like to ask the authors to comment on the following questions:

1) Regarding point 1: The proof for Theorem 2 mentions a relation to Markov compatibility by preserving topological structure. While hold true for sets $Z,Z^*$ is this also true for changing variables $D,D^*$? Could the authors elaborate this further, and comment on the above mentioned relation to faithfulness (which I believe is implied by Markov compatibility)?

2) Could the authors comment generally on the previous point 2? In particular, (1) would keeping $U_Z$ as parents of $Z,Z^*$ invalidate any of the presented results, and (2) would the application of rule 3 still be compatible for networks containing  $U$s that are shared between multiple variables?

3) Regarding point 3: Could the authors please provide more details on the proposed Multi-Scale Mixup Scheme (e.g. in the form of a pseudo algorithm)?

---

### Official Review · Reviewer_uSW2 · 2024-11-04

**Soundness:** 2
**Presentation:** 2
**Contribution:** 2
**Rating:** 3
**Confidence:** 3

**Summary:**

The paper proposed the teleporter theory for computing counterfactual queries. The theory mainly improves the original twin networks (or parallel networks) by merging certain duplicates into teleporters. The paper showed that the cross-world SCM constructed based on the teleporter theory may capture more independencies between variables in different worlds, leading to a more general adjustment formula for identifying counterfacutal queries under evidence.

**Strengths:**

Most parts of the paper are clearly written and easy to follow. Examples are provided for the definitions and theorems. Proofs of the theorems were provided in the appendix. The paper contains some empirical studies and applications. The paper also summarized some previous works and highlighted the improvements made by this work.

**Weaknesses:**

Here are my two major concerns:
1. While not mentioned, it seems that the paper assumes causal sufficiency (no hidden confounders) for all examples and Theorem 1 Rule 3. This can be a huge limitation when compared to the existing work.
2. The identification of counterfactual queries has been studied in [Shpitser & Pearl 2008] Section 4. This previous paper observed that the twin (parallel) networks may not capture the independencies and proposed "merge-cg" operation to merge variables in the twin (parallel) network. To prove the novelty, I think a thorough comparison between the method in [Shpitser & Pearl 2008] and the method in this paper is required.

Here are some issues regarding the formulation of theoretical results:

3. The notion of "cross-world SCM under teleporter theory" was never formally defined. In fact, I think Theorem 1 in its current form should really be a Definition not a Theorem. I'm also not sure what Theorem 1 is trying to prove here.
4. Similarly, in Theorem 2, d-separation has never been defined on cross-world SCMs under teleporter theory yet, so the notion is not well-defined. It's possible that the authors are unaware of the difference between independence and d-separation. d-separation is a notion defined only on DAGs, whereas (conditional) independencies are more generally defined for variables.
5. I found it difficult to penetrate Theorem 3. You may simply say "If a variable set $Z$ satisfies that $X$ and $Y_x$ are d-separated by $E \cup Z$, then the counterfactual query can be computed as follows: ..."

Comments on the experimental section:

6. The experimental section (Section 6) is quite confusing in general. I think the authors are stacking too many concepts together, e.g., the paragraph below Equation (6). I think the task/setup needs to be clarified further.
7. From my understanding, the goal of the experiments is to show that Equation (6) can lead to better counterfactual identification. If that's the case, I think the paper should explicitly state which methods in Table 1 actually consider Equation (6) and which ones don't. Also, I don't think methods based on neural networks are the best to demonstrate the effectiveness of Equation (6) -- the performance of neural networks can be quite arbitrary.

References:
- Ilya Shpitser, Judea Pearl. Complete Identification Methods for the Causal Hierarchy. J. Mach. Learn. Res. 9: 1941-1979 (2008).

**Questions:**

1. Section 5.2 paragraph 1. "However, obtaining the distribution of exogenous variables $U$ is extremely difficult". The term "difficult" is vague here. Suppose causal sufficiency is assumed in this paper, why is it difficult to learn the distribution of $U$ from data? You also need to compare Theorem 3 with the existing identification methods in [Shpitser & Pearl, 2008].

---

### Official Review · Reviewer_w7aC · 2024-11-07

**Soundness:** 3
**Presentation:** 3
**Contribution:** 1
**Rating:** 3
**Confidence:** 4

**Summary:**

This paper points out the issues with existing frameworks for counterfactual representations, such as twin networks or SWIG. The authors propose the teleporter theory, which offers a simpler graphical representation and algebraic derivation. Finally, they show the application of their proposed algorithm on the GraphOOD learning task, where the goal is to learn the invariant features of a graph.

**Strengths:**

The paper contains appropriate citations. The paper is quite well written. It discusses different topics intuitively. The application of counterfactuals in GraphOOD is quite interesting.

**Weaknesses:**

Below I provide my feedback. I hope these comments become useful to the authors.

## Minor weakness:
* The authors discussed many contributions/applications in the introduction. It might be a little hard to focus.
* Line 161: Since the authors are dealing with counterfactual variables, they should intuitively explain what $A \perp D_a | B$ or $A \perp D_a | C$ imply.
* Line 167: Theorem 4.3.1 and its intuitive explanation should be provided in the paper (maybe in the appendix). The authors can explicitly discuss how B, C satisfy the backdoor criterion for $(A, D)$.
* Line 200: The notation $Y(x)$ should be explained. The authors should also discuss why SWIG failed in such a scenario.

## Major weakness
* I **highly recommend** the authors check Shpitser (2008). They have already discussed the issues with twin networks (Page 1957, Figure 9, Figure 11): i) it being complicated, ii) d-separation criterion being misleading.
* Please check Lemma 24, Lemma 25, and Figure 10 (make-cg algorithm). They discuss how to detect which random variables are the same variables in the twin network and how to merge them. To my understanding, these lemmas and the algorithm do the same thing the proposed teleporter theory does in this paper. I would request the authors clearly specify their novelty compared to this work.

---
References:
* [shpitser2008complete] Shpitser, Ilya, and Judea Pearl. "Complete identification methods for the causal hierarchy." (2008). (link: https://www.jmlr.org/papers/volume9/shpitser08a/shpitser08a.pdf )
* Shpitser, Ilya, and Judea Pearl. "What counterfactuals can be tested." arXiv preprint arXiv:1206.5294 (2012).

**Questions:**

## Questions:
* To check whether $A \not\perp D_a | B$, why did we condition on $B$ but not $B_a$? Since $B$ and $B_a$ have the same set of parents and exogenous noise, aren't they the same variables? Should the authors condition on $B_a$ for determining such independency?
* How is $A \perp D_a | B$ different from $A \perp D_a | B_a$ or $A \perp D_a | [B, B_a]$? What would be the meaning of each such expression in terms of counterfactuals?
* What is the intuitive meaning of $P(D_0, A=1)$?
* What queries become counterfactually identifiable [1] that were non-identifiable before? Can the authors discuss on that?

[1] Shpitser, Ilya, and Judea Pearl. "Complete identification methods for the causal hierarchy." (2008).

---

### Note · Authors · 2024-11-13

I have read and agree with the venue's withdrawal policy on behalf of myself and my co-authors.